

# Quantum information and the C-theorem in de Sitter

Nicolás Abate and Gonzalo Torroba

Centro Atómico Bariloche and CONICET,
Instituto Balseiro, UNCuyo and CNEA,
S.C. de Bariloche, Río Negro, R8402AGP, Argentina

## Abstract

Information-theoretic methods have led to significant advances in nonperturbative quantum field theory in flat space. In this work, we show that these ideas can be generalized to field theories in a fixed de Sitter space. Focusing on 1+1-dimensional field theories, we derive a boosted strong subadditivity inequality in de Sitter, and show that it implies a C-theorem for renormalization group flows. Additionally, using the relative entropy, we establish a Lorentzian bound on the entanglement and thermal entropies for a field theory inside the static patch. Finally, we discuss possible connections with recent developments using unitarity methods.

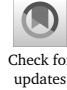

# 1 Introduction

Quantum information methods have led to key discoveries in nonperturbative quantum field theory. Of particular relevance to our work, the combination of strong subadditivity of the entanglement entropy with Poincare invariance implies the irreversibility of the renormalization group (RG) in $1+1$, $2+1$ and $3+1$ space-time dimensions, while also imposing constraints in higher dimensions [1–3]. However, much less is understood about theories without Poincare invariance. An important goal is to extend the results from quantum information theory to these more general settings, something that would be of direct relevance for condensed matter physics and cosmology.

In this work we take a step in this direction, by applying information-theoretic tools to quantum field theory (QFT) in a fixed de Sitter ($dS$) space-time. This obeys different motivations. First, the de Sitter universe, being the simplest and most symmetric cosmological space-time with positive cosmological constant, serves as a crucial testing ground. Understanding field theory dynamics in a fixed $dS$ background is a necessary step toward formulating quantum gravity in cosmological spacetimes.[1] Second, the vacuum correlators restricted to the static patch of $dS$ have a thermal interpretation, providing an opportunity to integrate information theory and finite temperature physics. Also, in $dS$ there is no globally conserved energy, and this offers a new setting for information theory and algebraic ideas [5,6].

Let us also highlight some more technical motivations. The success of information theory for field theories in flat space is due in part to the high degree of symmetry of Minkowski space-time. Since de Sitter space, like Minkowski, is maximally symmetric, this gives us hope that information theory measures can also give powerful probes for the dynamics in $dS$. Moreover, results from information theory in flat space QFTs reveal a close connection with correlator methods based on unitarity and causality [7–11], as well as with works on the F-theorem [12,13]. Recently, motivated in part by cosmology and quantum gravity, some of these tools have started to be developed in $dS$ as well, see e.g. [14–17]. This further motivates exploring how information theory methods might be extended to de Sitter space, and how they relate to these recent results.

We focus on QFTs that are obtained by perturbing a conformal field theory (CFT) with relevant scalar operators. The CFT acts as the UV fixed point, and the addition of relevant deformations triggers a nontrivial RG flow. It is possible to probe the RG flow by looking at observables at different distance scales. In flat space-time, one can consider distance scales much larger than those associated to the relevant couplings. We assume that this IR limit is described by a different CFT (which could be trivial if for instance there is a mass gap). This RG flow is said to be irreversible if it is possible to identify an "RG charge" that decreases between the UV and the IR.

The information-theoretic approach leverages the fact that the RG charges (e.g. $C, F, A$ in $d = 1+1, 2+1, 3+1$ dimensions) appear as specific terms in the entanglement entropy for the vacuum state reduced to a spherical spatial region. The goals are to isolate those terms and to establish their monotonicity properties under RG flows. It turns out that these goals are related: combining the strong subadditivity of the EE with Lorentz invariance of the vacuum (which allows to boost regions), one can derive a second order differential inequality for the EE. This inequality eliminates the nonuniversal terms, and lead to monotonicity properties of the desired universal terms [1–3]. A crucial aspect of this framework is that the EE of the UV fixed point saturates the differential inequality, a feature known as the Markov property of the conformal field theory vacuum. This fact allows to work with entropy differences or with relative entropies [18]. The information theory analysis has achieved a mathematical and

---

[1]As emphasized recently in [4], one-loop corrections to the de Sitter entropy can provide constraints on microscopic models of quantum gravity.

conceptual unification of irreversibility results, including those in odd space-time dimensions – something that has not been accomplished by other methods.

The de Sitter metric is conformally equivalent to the Minkowski one. As we will discuss below, this implies that the universal RG charges of CFTs can also be extracted from the EE for a spherical region in $dS$. While the conformal transformation relates the dynamics of UV fixed points in $dS$ and in flat space, the physics away from the UV limit is very different in both cases. In $dS$, the exponential expansion or contraction results in time-dependent couplings. A closely related challenge is that it is not possible to probe arbitrarily long distance scales while retaining causal contact. Alternatively, restricting to the (time-independent) static patch, there is a smallest energy scale set by the de Sitter temperature. These complications, which are absent in flat space, introduce challenges that we will have to address in our approach.

In this work, we will mostly address QFTs with nontrivial RG flows in 1+1-dimensional $dS$, postponing the higher dimensional case to future work [19]. After reviewing in Sec. 2 some basic properties, Sec. 3 explores the relative entropy as a distinguishability measure between the state associated to the UV fixed point and that along the flow. By taking an appropriate null limit for the Cauchy surface, the relative entropy is shown to agree with (minus) the entanglement entropy difference for the two states. Positivity of the relative entropy gives rise to a bound on both the entanglement and thermal entropies for a QFT in $dS$. Additionally, the monotonicity of the relative entropy results in a weak version of the C-theorem in $dS$. In Sec. 4 we study the interplay between strong subadditivity (SSA) and $dS$ isometries. By boosting causal diamonds, we arrive to a 'boosted SSA' inequality that is stronger than the purely spatial one. In Sec. 5 we apply these results to derive nonperturbative constraints on the QFT dynamics. We demonstrate that a conformal field theory (CFT) saturates the boosted SSA and thus leads to a Markov state; for a QFT, the boosted SSA is shown to imply the strong version of the C-theorem. We compare this with the situation in flat space, and with recent results using unitarity methods. Finally, we present our conclusions and discuss future directions in Sec. 6.

## 2  QFT and entanglement in dS

In this section, we describe the framework for our paper. Our main result, the C-theorem in $dS$, will apply to QFTs in $d = 1 + 1$ space-time dimensions, but it is useful to begin with considerations in general dimension $d$.

We consider a QFT in a fixed Lorentzian $d$-dimensional de Sitter space, $dS_d$. This spacetime is defined as a hyperboloid embedded in $\mathbb{R}^{1,d}$,

$$-(X^0)^2 + (X^d)^2 + (X^i)^2 = \ell^2, \quad i = 1, \ldots, d-1, \tag{1}$$

where $\ell$ is the $dS$ radius. We will mostly work with the global conformal coordinates, which are given by

$$X^0 = \ell \tan T, \quad X^d = \ell \frac{\cos \theta}{\cos T}, \quad X^i = \ell \frac{\sin \theta}{\cos T} \hat{x}^i, \quad i = 1, \ldots, d-1, \tag{2}$$

where $\hat{x}^i$ is a unit vector describing the sphere $S^{d-2}$. The coordinate ranges are

$$-\frac{\pi}{2} \leq T \leq \frac{\pi}{2}, \quad 0 \leq \theta \leq \pi, \tag{3}$$

for $d \geq 3$. In $1 + 1$ dimensions, the angular range is

$$-\pi \leq \theta \leq \pi, \quad (d = 1 + 1). \tag{4}$$

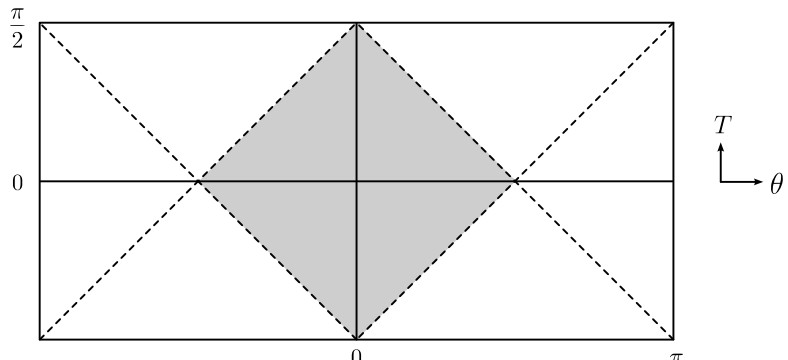

Figure 1: Unwrapped Penrose diagram of $dS_d$. The shaded area is the static patch, which is the region causally connected to the trajectory of a static observer. The left and right edges are identified.

Using the coordinates (2), the $dS$ metric reads

$$ds_d^2 = \frac{\ell^2}{\cos^2 T}\left(-dT^2 + d\theta^2 + \sin^2\theta\, d\Omega_{d-2}^2\right). \tag{5}$$

These coordinates cover all of $dS_d$ and make explicit the causal structure. They define the Penrose diagram of the spacetime, shown in Fig. 1.

The connected isometry group of $dS_d$ is $SO(1,d)$. It can be obtained by restricting the Lorentz generators of the embedding space,

$$J_{MN} = X_M \partial_N - X_N \partial_M, \tag{6}$$

to the hyperboloid. These transformations will play an important role in our application of quantum information methods. Although both de Sitter and Minkowski space-time are maximally symmetric, a key difference is that $dS$ has no globally conserved time-like Killing vector.

However, it is possible to restrict to part of the space-time, the static patch, where there is a conserved time-like Killing vector. The static patch is defined by

$$X^0 = \ell\cos\psi\sinh t, \quad X^d = \ell\cos\psi\cosh t, \quad X^i = \ell\sin\psi\,\hat{x}^i, \tag{7}$$

with coordinate ranges

$$-\infty < t < \infty, \quad -\pi/2 \le \psi \le \pi/2. \tag{8}$$

The metric becomes[2]

$$ds_d^2 = \ell^2\left[-\cos^2\psi\, dt^2 + d\psi^2 + \sin^2\psi\, d\Omega_{d-2}^2\right]. \tag{9}$$

The static patch describes the causal diamond for an observer at $\psi = 0$ (namely $\theta = 0$ or $\theta = \pi$), in the global coordinates. $\psi = \pm\pi/2$ defines the horizon associated to the causally accessible region. The static patch is shown in Fig. 1.

Our general goal is to understand the dynamics of quantum field theory in $dS_d$. We start from a conformal field theory at short distances, and the QFT is defined by perturbing the action by relevant primary scalar operators $\phi_I$ of scaling dimension $\Delta_I < d$,

$$S_{\text{QFT}} = S_{\text{CFT}} + \int d^d x\,\sqrt{-g}\,\lambda_I\phi_I. \tag{10}$$

---

[2]In the literature it is also customary to work with the coordinate $\rho = \sin\psi$, in terms of which the metric becomes

$$ds_d^2 = \ell^2\left[-(1-\rho^2)dt^2 + (1-\rho^2)^{-1}d\rho^2 + \rho^2 d\Omega_{d-2}^2\right].$$

However, to avoid confusions we stick to $\psi$ since in our work the symbol $\rho$ will be extensively used to denote density matrices.

The CFT need not to be described by an action, and (10) means that we are inserting the exponential of the relevant deformations in the path integral (or in all correlation calculations). At distance scales $\Delta x$ short compared to $\lambda_I^{-1/(d-\Delta_I)}$, the correlation functions are well approximated by that of the UV CFT, but once $\Delta x \gtrsim \lambda_I^{-1/(d-\Delta_I)}$ the relevant couplings induce significant deformations. The theory becomes strongly coupled, requiring nonperturbative methods in order to understand its dynamics.

We wish to use quantum information methods to characterize nonperturbatively such RG flows. This question is much more nontrivial than in Minkowski space-time. The metric is time-dependent and this induces time-dependent couplings along the flow. Furthermore, the de Sitter radius $\ell$ introduces another scale in the problem, and the dynamics depends also on dimensionless combinations $\lambda_I^{1/(d-\Delta_I)}\ell$. These effects are not present in flat space-time.

The euclidean version corresponds to placing the QFT on a sphere of radius $\ell$. Since Cardy's conjecture [20], understanding the nonperturbative dynamics of field theories on the sphere has been a long-standing objective. At fixed points, the free energy on the sphere determines intrinsic universal quantities, for which the irreversibility of flat-space RG flows in $1+1$, $2+1$ and $3+1$ space-time dimensions has been established [2,3,7,9]. The radius $\ell$ provides a natural scale to probe renormalization group flows. The literature on this subject is vast, and we just note [17,21], which provided direct motivation for our work. However, varying $\ell$ does not directly access different RG scales within the same theory; instead, it changes the quantum field theory. This makes it harder to apply algebraic and information-theoretic methods. Our approach here will be different: we keep $\ell$ fixed, and compute quantum information measures in subregions of the space-time.

## 2.1 Vacuum and entanglement

We will consider the QFT on the Euclidean vacuum, which we denote by $|E\rangle$ [22–25]. In this state, the correlation functions are obtained by analytic continuation from the sphere $S^d$. This state can be prepared by performing the euclidean path integral between $\tau = -\infty$ and $\tau = 0$, without insertions, where $\tau = iT$.[3] The Euclidean vacuum is invariant under the $dS$ isometries but, unlike the Minkowski vacuum, it does not arise as a state of minimum energy (there is no positive conserved energy).

Restricting the pure state $|E\rangle$ to the static patch (9) gives a mixed thermal density matrix; the local conserved Hamiltonian generates time translations $J_{0d} = \partial_t$ inside the static patch. This is analogous to restricting the Minkowski vacuum to the Rindler wedge, which is thermal with respect to the boost generator that preserves the wedge. See e.g. [5,6] for a discussion of these points in the algebraic formulation of QFT.

We will be interested in evaluating information theory measures, such as the entanglement and relative entropy, associated to different subregions of space-time. Different possibilities are shown in Fig. 2.

The simplest one is a causal diamond on the $T = 0$ Cauchy surface, inside the static patch. This is shown in blue in the figure. The entanglement and relative entropies for this region will provide a C-function that we will use to establish the irreversibility of RG flows. Let's denote the range of $\theta$ for this region by $0 \leq \theta \leq \theta_0$ ($|\theta| \leq \theta_0$ for $d = 1+1$). Recalling the metric (5),

---

[3]The infinite past in the euclidean metric

$$ds_d^2 = \frac{\ell^2}{\cosh^2 \tau} \left( d\tau^2 + d\theta^2 + \sin^2 \theta \, d\Omega_{d-2}^2 \right),$$

corresponds to a vanishing volume (the southern pole of the euclidean sphere), and the relevant deformations do not contribute in this limit.

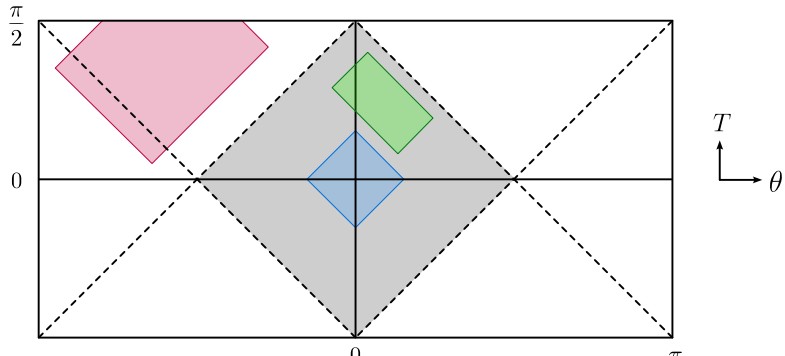

Figure 2: Three examples of causal developments in $dS$. The blue diamond corresponds to an entangling region at $T = 0$ that fits inside the static patch. Applying $dS$ isometries to such a region leads to more general diamonds, such as the green one. Other causal developments correspond to truncated diamonds, such as the one in pink, whose causal complement is also a truncated diamond. We will not consider such regions in this work.

the entangling region associated to this causal diamond is given by a sphere of radius

$$R = \ell \sin\theta_0 \,. \tag{11}$$

The field theory restricted to this region is described by a density matrix, obtained as a partial trace from the thermal density matrix of the static patch, or from the pure Euclidean vacuum. For $\theta_0 = \pi/2$, we have $R = \ell$, the density matrix becomes thermal with respect to the static patch time, and the entanglement entropy agrees with the thermal entropy of $dS$. The causal development for the region at $T = 0$ with $\theta_0 > \pi/2$ is a truncated diamond. It can be described in terms of its complement, which is a causal diamond at $T = 0$ that fits inside the static patch centered at $\theta = \pi$. The algebra of operators can also be defined as the commutant of the algebra of the complementary region.

In our construction, we will need to consider causal diamonds obtained by applying $dS$ isometries to the diamond at $T = 0$ that we just described. An example is shown in green in Fig. 2. This will be required in Sec. 4 in order to derive a boosted strong subadditivity inequality. A third kind of causal development in $dS$ is a truncated diamond whose complement is also a truncated diamond. An example is shown in the pink region in the figure above. This corresponds to entanglement in super-horizon scales, and leads to a time-dependent entanglement entropy; see for instance [26]. We will not consider such regions in this paper, but it would be interesting to understand their properties in more detail.

## 2.2 Structure of the entanglement entropy

As in flat space, the vacuum entanglement entropy (EE)

$$S(\rho_V) = -\text{tr}_V \, \rho_V \log \rho_V \,, \tag{12}$$

for a region $V$ in $dS$ is divergent due to UV contributions. Here we will instead be interested in the entropy difference between the quantum field theory that undergoes the RG flow (whose density matrix we denote by $\rho$) and that of the UV fixed point (with density matrix $\sigma$):

$$\Delta S(V) = S(\rho_V) - S(\sigma_V) \,. \tag{13}$$

In general dimensions, it is possible to supplement the UV fixed point with appropriate counterterms in order to render $\Delta S(V)$ finite.[4] Furthermore, we will see shortly that $\Delta S(V)$ has a natural interpretation in terms of the relative entropy.

In $d$ dimensions, the EE has divergent terms proportional to geometric invariants of the entangling regions, such as the area, curvature, etc. Let us focus on the universal term, which is also the leading contribution in $1+1$ dimensions. In even $d$, the entanglement entropy for a sphere of radius $R_0$ in Minkowski is related to the $A$-anomaly [29],

$$S_{\text{univ}}(R_0) = (-1)^{d/2-1} 4A \log(R_0/\epsilon), \tag{14}$$

where $\epsilon$ is a short distance cutoff. Our goal is to find the corresponding universal term for the EE in $dS_d$.

We perform a conformal transformation from a causal diamond of radius $\ell$ in Minkowski to a static patch of $dS_d$ with curvature radius $\ell$ [29, 30],

$$x^0 = \ell \frac{\cos\psi \, \sinh(t/\ell)}{1 + \cos\psi \, \cosh(t/\ell)}, \quad x^i = r\hat{n}^i = \ell \frac{\sin\psi}{1 + \cos\psi \, \cosh(t/\ell)} \hat{n}^i. \tag{15}$$

It relates the Minkowski and static patch space-times by a Weyl factor (whose explicit form won't be needed),

$$-(\mathrm{d}x^0)^2 + \mathrm{d}r^2 + r^2 \mathrm{d}\Omega_{d-2}^2 = \Omega_{dS}(t, \psi)^2 \left( -\cos^2\psi \, \mathrm{d}t^2 + \ell^2 \mathrm{d}\psi^2 + \ell^2 \sin^2\psi \, \mathrm{d}\Omega_{d-2}^2 \right). \tag{16}$$

We note that at $t = 0$ the static patch angle $\psi$ coincides with the global conformal coordinate $\theta$. A sphere of radius $R_0$ in Minkowski space-time at $x^0 = 0$ is then mapped to a sphere in de Sitter at $t = 0$ and radius

$$\ell \sin\theta_0 = \frac{2R_0}{1 + R_0^2/\ell^2}, \qquad R_0 = \ell \frac{\sin\theta_0}{1 + \cos\theta_0}. \tag{17}$$

Since the transformation is conformal, it maps null lines into null lines. Therefore, the causal development of the Minkowski sphere $\mathcal{D}_{R_0}^{(\text{Mink})}$ is mapped to the causal development of the $dS$ sphere $\mathcal{D}_{\theta_0}^{(\text{dS})}$.

Under the previous conformal transformation, the sphere in Minkowski maps to a sphere in $dS$ of radius $\ell \sin\theta_0$. We also have to map the short distance cutoffs. In flat space, the cutoff $\epsilon$ implies that the radius of the sphere is decreased to $R_0 - \epsilon$. In $dS$, the spatial coordinate that defines the sphere is the angle $\theta$, and so we should have an angular cutoff $\epsilon_\theta$. Applying the map from the cutoff sphere in Minkowski to the cutoff sphere in $dS$, we have

$$R_0 - \epsilon = \ell \frac{\sin(\theta_0 - \epsilon_\theta)}{1 + \cos(\theta_0 - \epsilon_\theta)}. \tag{18}$$

Using the relation between $R_0$ and $\theta_0$, and expanding for small $\epsilon_\theta$, we find

$$\epsilon \approx \frac{\ell \, \epsilon_\theta}{1 + \cos\theta_0}. \tag{19}$$

Therefore, the universal term of the EE in $dS$ becomes

$$S_{\text{univ}}(\theta_0) = (-1)^{d/2-1} 4A \log(\sin\theta_0/\epsilon_\theta). \tag{20}$$

---

[4]This is done by renormalizing the effective gravitational action produced by the QFT. The best known example is the divergence in the area term, which corresponds to the renormalization of the Newton's constant. See [27,28] and references therein.

In two dimensions, the entropy is logarithmically divergent, and (20) is the leading contribution for a conformal fixed point. The prefactor is $C/3$ in terms of the central charge, and the EE for a CFT in a spatial region inside the static patch of de Sitter is

$$S_{\text{CFT}}(\theta_0) = \frac{C}{3} \log\left( \frac{\sin(\theta_0)}{\epsilon_\theta} \right). \tag{21}$$

In particular, the difference between the EE of the theory with the RG flow and that of the UV fixed point is

$$\Delta S(\theta_0) = S_\rho(\theta_0) - \frac{C_{\text{UV}}}{3} \log\left( \frac{\sin(\theta_0)}{\epsilon_\theta} \right). \tag{22}$$

This entropy difference is free of UV divergences. The UV limit corresponds to $\theta_0 \to 0$, in which case $\Delta S \to 0$. We will also see in the next section that $\Delta S \leq 0$.

For a QFT in Minkowski space-time, it is possible to take an IR limit where the sphere radius is much larger than the inverse of the mass scale $m$ of the RG. However, this is not possible in $dS$, where the maximum size we can consider in our framework is $R = \ell$, corresponding to $\theta_0 = \pi/2$. Indeed, the vacuum state is pure, so the EE for a spatial region on the $T = 0$ Cauchy slice has to be equal to the entropy of the complement,

$$\Delta S(\theta_0) = \Delta S(\pi - \theta_0). \tag{23}$$

Therefore, $\Delta S \to 0$ for $\theta_0 = 0, \pi$, and it attains an extremum at $\theta_0 = \pi/2$. We will show in (43) and (46) that the first derivative of $\Delta S$ is negative for $\theta_0 < \pi/2$ (and positive for $\theta_0 > \pi/2$), so the extremum is in fact its minimum negative value. The entropy difference $\Delta S(\theta_0 = \pi/2)$ does not probe the IR fixed point in general, and depends on the RG flow. It coincides with the difference of thermal entropies in the static patch. We will study the physical meaning of this quantity below. We should contrast this with RG flows in Minkowski space-time. The IR Minkowski limit can be accessed in $dS$ by sending $\ell/R \to \infty$ and $m\ell \to \infty$ with $mR$ fixed, and then taking large $R$. We instead work at fixed $\ell$, and we will derive information-theoretic inequalities that involve varying $R$.

## 3 Relative entropy and the weak C-theorem

We begin our analysis of information-theoretic measures in $dS$ with a study of the relative entropy. Using its positivity and monotonicity, we will obtain a nontrivial bound on the entropy for $dS$, and our first weak version of the C-theorem for RG flows.[5]

The relative entropy is defined as

$$S_{\text{rel}}(\rho|\sigma) = \text{tr}(\rho \log \rho - \rho \log \sigma), \tag{24}$$

for two density matrices $\rho$ and $\sigma$. As we discussed in Sec. 2.2, we choose $\sigma$ as the vacumm state of the UV CFT in 1+1-dimensional $dS$ reduced to the causal development of the segment

$$\theta \in [-\theta_0, \theta_0], \quad T = 0, \tag{25}$$

while $\rho$ will be the state associated with the QFT that undergoes the RG flow triggered by the relevant deformation (10).

The relative entropy is positive and monotonic under inclusion of algebras. It measures the distinguishability between two states and, unlike the entanglement entropy, is well-defined for the continuum theory [33].

---

[5]The relative entropy is also directly relevant for the study of RG flows in flat space, see e.g. [31,32].

In order to compute the relative entropy between $\rho$ and $\sigma$, they need to be states in the same Hilbert space. In the presence of a lattice cutoff, one starts with degrees of freedom at each lattice point and then adds interactions between them. The difference between $\rho$ and $\sigma$ lies in the interactions, and hence the Hilbert space constructed by taking tensor products of the Hilbert spaces of each lattice point are the same (more precisely, there is an isomorphism once a Cauchy slice is chosen). Sec. 2 of [32] defined an isomorphism between the algebras of operators of the two theories directly in the continuum, which is closely related to the standard interaction picture in quantum field theory. This isomorphism allows to compute the relative entropy within the same theory (see e.g. Eq. (2.11) in [32]). A crucial feature of this construction is that the relative entropy depends on the choice of Cauchy surface (which enters into the isomorphism map). This dependence will play a significant role in what follows.

### 3.1 Null limit for the relative entropy

It is convenient to express the relative entropy as

$$S_{\text{rel}}(\theta_0) = \Delta\langle H \rangle - \Delta S \,, \tag{26}$$

where $\Delta\langle H \rangle = \text{tr}(\rho H) - \text{tr}(\sigma H)$ is the difference of the expectation values of the CFT modular Hamiltonian

$$H = -\log\sigma \,, \tag{27}$$

and $\Delta S = S(\rho) - S(\sigma)$ as defined at (13). Our goal is to relate the entropy difference $\Delta S$ to the relative entropy. However, in general the relative entropy is dominated by the modular Hamiltonian term, which scales like the space-time volume of the causal development, whereas the entropy scales like the spatial area.

An important property emphasized in [31,32] is that the modular Hamiltonian contribution to the relative entropy depends on the Cauchy surface over which the expectation values are being computed. The reason is that we are comparing states that evolve with different unitary operators; in order to evaluate $\text{tr}(\rho H)$ one has to identify the algebras of operators for $\rho$ and $\sigma$, and this is accomplished using some Cauchy surface.

Following [31,32], we will use the dependence on the Cauchy surface in our favor since in the null limit the modular Hamiltonian contribution vanishes. To demonstrate this we will need an expression for the modular Hamiltonian of the reduced vacuum state for a CFT in $dS_2$, which can be obtained by mapping the corresponding one in Minkowski spacetime with the conformal transformation (15). This yields

$$H = 2\pi \int_\Sigma \eta^\mu \xi^\nu T_{\mu\nu} \,, \tag{28}$$

where $T_{\mu\nu}$ is the energy momentum tensor of the CFT, $\Sigma$ is a Cauchy surface with future-pointing normal vector $\eta^\mu$, and

$$\xi^\nu = \left( \frac{\cos T \cos\theta}{\sin\theta_0} - \cot\theta_0, -\frac{\sin T \sin\theta}{\sin\theta_0} \right) \,. \tag{29}$$

This is a conformal Killing vector that preserves the causal development of the segment (25), as we show in Fig. 3. It coincides with the expression found in [34]. When $\theta_0 = \pi/2$, $\xi^\mu$ becomes the time-like Killing vector $\partial_t$ for the static patch, expressed in global coordinates.[6]

---

[6]This modular Hamiltonian for diamonds inside the static patch of de Sitter was also recently derived in [35,36].

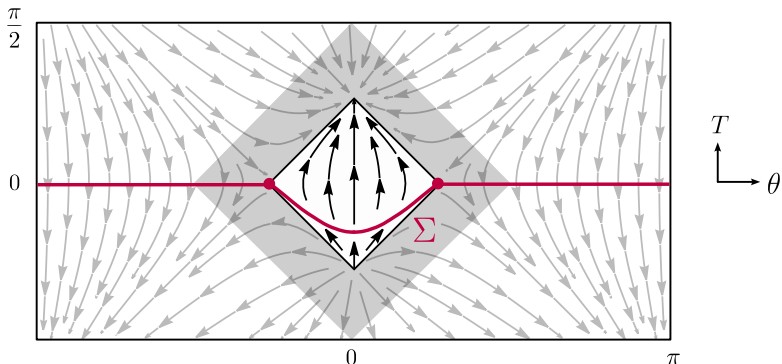

Figure 3: Conformal Killing vector (29) that preserves the causal diamond of the segment. In pink we also show the Cauchy surface (33) that approaches the past null boundary of the diamond for $a \to 0$.

The difference $\Delta\langle H \rangle$ is then determined by $\Delta\langle T_{\mu\nu} \rangle$, and this quantity only depends on local, symmetric tensors on the Cauchy surface. These are constructed using the normal $\eta_\mu$, the metric $g_{\mu\nu}$ and intrinsic and extrinsic curvatures. The contributions from curvatures are subleading because they are suppressed by the short distance cutoff. The leading contribution is then of the form

$$\Delta\langle T_{\mu\nu} \rangle = k_1 \eta_\mu \eta_\nu + k_2 g_{\mu\nu}, \tag{30}$$

in terms of constants $k_1$ and $k_2$. Then

$$\Delta\langle H \rangle = \pi k \int_\Sigma \eta^\mu \xi_\mu, \tag{31}$$

with $k$ a linear combination of $k_1$ and $k_2$. Since we are dealing with the CFT stress tensor, the first nontrivial perturbative correction to its expectation value must come at order $\lambda_I^2$. This is because one has $\langle T_{\mu\nu} \mathcal{O}_I \rangle = 0$ for any primary operator $\mathcal{O}_I$. Also, by dimensional analysis we expect that in terms of the regulator $\epsilon_\theta$ one has

$$k \sim \lambda_I^2 \epsilon_\theta^{2-2\Delta_I}, \tag{32}$$

so in general $\Delta\langle H \rangle$ will be finite only for deformations with $\Delta_I < 1$.[7]

Now let us evaluate (31) for a Cauchy surface in the past light cone of the entangling region. We can take the null limit for example by considering the following hyperboloid

$$\Sigma = \left\{ \left(T + \sqrt{a^2 + \theta_0^2}\right)^2 - \theta^2 = a^2, \quad |\theta| \le \theta_0, \ T < 0 \right\}, \tag{33}$$

that for $a \to 0$ approaches the curve $T = |\theta| - \theta_0$, i.e. the past null boundary of the causal diamond associated with our region. Writing the defining equation of $\Sigma$ as $\Phi(T,\theta) = 0$, the normal is given by $\eta^\mu \propto \partial^\mu \Phi$ (and a suitable normalization), so

$$\int_\Sigma \eta^\mu \xi_\mu = \int_{\epsilon_\theta}^{\theta_0} d\theta \, \frac{\ell^2}{\cos(\theta - \theta_0)} \left[ \left( \frac{\theta - \theta_0}{\theta \theta_0} + \frac{\sin\theta \sin(\theta - \theta_0)}{\theta^2 \sin\theta_0} \right) \right] a^2 + \mathcal{O}(a^3). \tag{34}$$

We have introduced the cutoff $\epsilon_\theta$ in order to regularize the integral, allowing us to expand in powers of $a$. The remaining integral is bounded by $\log(\epsilon_\theta)$, so taking $\epsilon_\theta \sim a \to 0$ we see that $\Delta\langle H \rangle \sim a^2 \log a$ vanishes. Moreover, recalling (32), this also means that the convergence window for $\Delta\langle H \rangle$ is enhanced to $\Delta_I < 2$, i.e. we can safely consider any relevant deformation in the null limit.

---

[7]One could also include a dependence on $\ell$ in (32). However, near the UV fixed point the radius can only appear with negative powers. Thus this contribution is subleading in $\epsilon_\theta$ with respect to the term considered in (32).

## 3.2 Bound on entanglement entropy

Let us now study the consequences of $\Delta\langle H\rangle|_{\text{null}} = 0$. This implies that in the null limit

$$S_{\text{rel}}(\theta_0) = -\Delta S(\theta_0),\tag{35}$$

and hence the entropy difference has an interpretation as the relative entropy between the CFT and QFT density matrices. This is a well-defined information measure in the continuum theory.

Positivity of $S_{\text{rel}}$ gives a bound for the entanglement entropy difference. Recalling (22), we have that

$$\Delta S \le 0, \quad\text{or}\quad S_\rho(\theta_0) + \frac{C_{\text{UV}}}{3}\log(\epsilon_\theta) \le \frac{C_{\text{UV}}}{3}\log(\sin\theta_0).\tag{36}$$

The $\log(\epsilon_\theta)$ in the left hand side of the second inequality cancels the UV divergence of the entanglement entropy, giving a finite regularized entropy. We see that this is upper-bounded by the UV central charge.

When $\theta_0 = \pi/2$, the entanglement entropy becomes the thermal entropy of the QFT in the static patch, and the relative entropy becomes the difference of free energies. Therefore, we obtain the following equality with a thermodynamic interpretation,

$$\Delta S_{\text{thermal}} = -S_{\text{rel}}(R = \ell).\tag{37}$$

We stress that the setup for establishing this relation is intrinsically Lorentzian: it requires working with a Cauchy slice that approaches the past light-cone, in order to set to zero the modular Hamiltonian contribution. Curiously, (37) also appears in the presence of dynamical gravity, but due to different reasons (related to the Hamiltonian constraint in $dS$); see e.g. [37]. Positivity of the relative entropy then leads to an interesting bound on the maximum thermal entropy for a QFT in the static patch of $dS$:

$$S_{\text{thermal}} + \frac{C_{\text{UV}}}{3}\log(\epsilon_\theta) \le 0.\tag{38}$$

Here the thermal entropy is assumed to be regualirzed using the same short distance cutoff $\epsilon_\theta$ that appears in the EE. Then the second term in the left hand side of (38) subtracts the UV divergence from the thermal entropy. There is no analog of this thermal inequality for a QFT in Minkowski space-time, where there is no local thermal interpretation for bounded entangling regions.

## 3.3 The C-theorem: weak version

In order to prove the C-theorem in de Sitter we will use the monotonicity of the relative entropy. This will give a weak version, in the sense of fixing the sign for a running C-function, but not its first derivative. The strong version will be obtained using boosted strong subadditivitiy in Sec. 4.

The relative entropy is monotonic under inclusion of algebras. Let us consider the algebra associated to the causal development of the region defined by $T = 0$ and $-\theta_0 \le \theta \le \theta_0$, and that associated to a slightly smaller causal diamond. We need to evaluate the relative entropy on the same state which, as we explained in the previous subsection, we defined on a null Cauchy surface.[8] Therefore, the two causal diamonds have to share the same past light-cone. We show this in Fig. 4. The Euclidean vacuum is invariant under the $dS$ isometries, and so the relative entropy is a function of the geodesic distance between the spatial endpoints of the causal diamonds.

---

[8]The modular Hamiltonian of the smaller diamond is given by a conformal Killing vector that generalizes (29), which can be obtained using a boost $J_{02}$. However, its contribution on the null surface also vanishes, so we can safely consider $S_{\text{rel}} = -\Delta S$ on this smaller region as well.

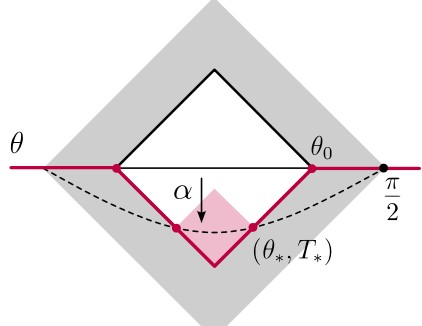

Figure 4: In order to evaluate the relative entropy on the same state defined on the null Cauchy surface $T = |\theta| - \theta_0$ (the pink curve on the figure), we apply a boost $J_{02}$ of parameter $\alpha$ to the $T = 0$ slice. This defines a new diamond – whose spatial endpoints we denote by $(\pm\theta_*, T_*)$ – which is contained in and shares the past light cone with the original one. Monotonicity of the relative entropy then implies that this magnitude decreases with $\alpha$.

We denote the spatial endpoints of the new causal diamond by $(\pm\theta_*, T_*)$, with $T_* = \theta_* - \theta_0$. As reviewed in the next section, the space-like geodesic joining these two points can be obtained from a boost of the original geodesic at $T = 0$. Introducing the boost parameter $\alpha$ defined as

$$\frac{\sin T_*}{\cos \theta_*} = \frac{2 \sin \theta_0}{1 + \coth \alpha}, \tag{39}$$

a short calculation gives the geodesic length

$$L(\alpha) = 2\ell \arctan\left(\frac{(\coth \alpha - 1)\tan \theta_0}{\sqrt{(1 + \coth \alpha)^2 - 4 \sin^2 \theta_0}}\right). \tag{40}$$

For $\alpha = 0$ we recover the length of the spatial geodesic between $(T = 0, \theta = \pm\theta_0)$,

$$L = 2\ell\, \theta_0. \tag{41}$$

Monotonicity of the relative entropy then implies that

$$S_{\rm rel}(L) \geq S_{\rm rel}(L(\alpha)). \tag{42}$$

In the limit $\alpha \to 0$, this leads to the differential inequality

$$\sin(2\theta_0)\, \partial_{\theta_0} S_{\rm rel}(\theta_0) \geq 0. \tag{43}$$

For a CFT, the central charge $C$ is related to the EE (21) by

$$C = 3 \tan \theta_0\, \partial_{\theta_0} S_{\rm CFT}(\theta_0). \tag{44}$$

This motivates the definition of an entropic running C-function

$$C(\theta_0) = 3 \tan \theta_0\, \partial_{\theta_0} S(\theta_0), \tag{45}$$

which satisfies $C(\theta_0 \to 0) = C_{\rm UV}$. From (43) and recalling $\theta_0 \leq \pi/2$, we conclude that

$$\Delta C = 3 \tan \theta_0\, \partial_{\theta_0} \Delta S(\theta_0) \leq 0, \quad \text{or} \quad C(\theta_0) \leq C_{\rm UV}. \tag{46}$$

Note that this inequality actually implies (36). It also means that the RG flow is irreversible in the sense that the function $C$ at any finite scale $\theta_0 \neq 0$ is always smaller than the UV central charge – which happens to coincide with $C(\theta_0 \to 0)$. We also learn that the irreversibility is related to the distinguishability between the reduced states of the QFT and the UV CFT at finite scales, provided that we arrived at (46) using the relative entropy between these states. However, this inequality is not sufficient to show that the function $C$ is monotonic along the RG flow, so it is in this sense that we call it the weak C-theorem in de Sitter. The monotonicity of $C$ will be derived in the next sections.

## 4 Boosted strong subadditivity in de Sitter

A key idea in the application of quantum information methods to field theory was to combine strong subadditivity with Lorentz invariance [1]. The resulting boosted SSA leads (after years of work) to the C, F and A theorems for irreversibility of RG flows [1–3]. In this section we will show that it is also possible to combine the SSA with $dS$ invariance in order to derive a differential inequality that leads to the irreversibility of RG flows in de Sitter. We focus on 1+1-dimensional theories, but preliminary results suggest that these methods can be generalized to higher dimensions [19].

We begin with the causal diamond associated with the segment (25), and consider more general regions obtained from it by applying $dS$ isometries. First, let us perform a boost $J_{02}$ of the segment, by a boost angle $\alpha$. Combining this with a rotation $J_{12}$ of angle $\theta_0$, we can keep the left-point $\theta = -\theta_0$ fixed, and require the right-point to lie on the boundary of the causal diamond of the segment. As we show in the left panel of Fig. 5, this procedure defines a new geodesic curve on $dS$ which we denote by $A$. Its parametrization in global coordinates is given by

$$\sin T = \operatorname{th} \alpha \sin(\theta + \theta_0), \quad \theta \in (-\theta_0, \theta_*). \tag{47}$$

Here $\theta_*$ lies at the intersection with the light-cone $T = \theta_0 - \theta$, namely

$$\sin(\theta_0 - \theta_*) = \operatorname{th} \alpha \sin(\theta_0 + \theta_*) \quad \Rightarrow \quad \tan \theta_* = \tan(2\theta_0) \frac{1 - \operatorname{th} \alpha}{1 + \operatorname{th} \alpha}. \tag{48}$$

The curve (47) is a space-like geodesic of $dS$; it can be obtained as the intersection of a (boosted) plane through the origin with the hyperboloid in the embedding space. Its geodesic length computed using the line element (5) is given by

$$L_A = \ell \arctan\left(\frac{1}{\cosh \alpha} \frac{\sin(2\theta_0)}{\operatorname{th} \alpha + \cos(2\theta_0)}\right). \tag{49}$$

Let us also perform the symmetric operation on the segment (25), but this time keeping the right-point $\theta = \theta_0$ fixed and boosting such that the left-point lies on $T = \theta_0 + \theta$. We denote the resulting geodesic by $B$, which has the same length as $A$.

Note that the starting causal diamond is simply $A \cup B$ in the causal sense. Its geodesic length is

$$L_{A \cup B} \equiv L = 2\ell \, \theta_0. \tag{50}$$

The causal intersection $A \cap B$ is given by the causal development of the geodesic with end-points $(\pm \theta_*, T_*)$, where $T_* = \theta_0 - \theta_*$. This geodesic simply corresponds to a time-evolution of the original segment (25) along the static patch, or a $J_{02}$ boost of the global coordinates by parameter $\tilde{\alpha}$. It is given by

$$\sin T = \operatorname{th} \tilde{\alpha} \cos \theta, \tag{51}$$

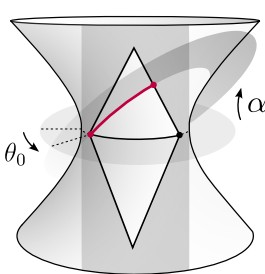
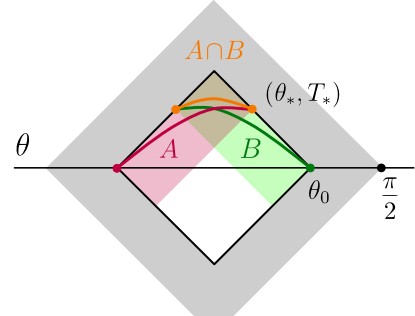

Figure 5: Left: construction of the curve (47) applying $dS$ isometries to the segment (25), namely a boost $J_{02}$ of angle $\alpha$ combined with a rotation $J_{12}$ of angle $\theta_0$. Both the segment (black) and the new curve (pink) are intersections of the hyperboloid with some plane trough the origin, and thus are geodesics of $dS$. Right: geodesics and causal diamonds involved in the strong subadditivity inequality (54) on the Penrose diagram. In both panels, the static patch is displayed in gray and the causal development of the segment in white.

where the boost angle is fixed by the endpoints:

$$\operatorname{th}\tilde{\alpha} = \frac{\sin T_*}{\cos\theta_*} = \frac{2\sin\theta_0}{1+\coth\alpha}\,. \tag{52}$$

The resulting geodesic length is then

$$L_{A\cap B} = 2\ell \arctan\left(\frac{(\coth\alpha - 1)\tan\theta_0}{\sqrt{(1+\coth\alpha)^2 - 4\sin^2\theta_0}}\right)\,. \tag{53}$$

Let us apply the construction we have presented so far, which we show in the right panel of Fig. 5, to the strong subadditiviy of the entanglement entropy

$$S_A + S_B \geq S_{A\cup B} + S_{A\cap B}\,. \tag{54}$$

Since the vacuum state is de Sitter invariant, the entanglement entropy associated to a region can only depend on the geodesic distance between its endpoints. Thus (54) for the boosted diamonds, and their union and intersection, becomes

$$2S(L_A) - S(L) - S(L_{A\cap B}) \geq 0\,. \tag{55}$$

Replacing (49) and (53) into (55) gives a finite boosted SSA inequality. Taking the limit $\alpha \to 0$, this gives rise to the differential version of boosted SSA:

$$\frac{1}{2}\sin(2\theta_0)\,\partial_{\theta_0}^2 S(\theta_0) + \partial_{\theta_0} S(\theta_0) \leq 0\,. \tag{56}$$

This boosted SSA inequality for de Sitter is one of our main results.

## 5  The C-theorem: Strong version

The setup in Fig. 5 of boosted causal diamonds was used to prove the C-theorem in flat space in [1]. A new element in de Sitter is the dependence on proper lengths that are sensitive to the curved metric of the space-time. We wish to understand the implications of this for information-theoretic inequalities for QFTs in de Sitter, and especially the consequences for the irreversibility of renormalization group flows.

## 5.1 Markov property and proof of the strong C-theorem

We first note that the EE for a CFT, eq. (21), saturates the SSA inequality (56), or equivalently,

$$S_{\text{CFT}}(A) + S_{\text{CFT}}(B) = S_{\text{CFT}}(A \cap B) + S_{\text{CFT}}(A \cup B), \tag{57}$$

for regions with boundary on the causal diamond. The saturation of the SSA leads to a Markov state [38]. It was shown in [18] that the CFT vacuum state in flat space is Markovian for regions with boundary on the light-cone. Here we see that the same holds for a CFT in de Sitter in $1 + 1$-dimensions. We will generalize this property to higher dimensions in [19].

The EE satisfies the SSA inequality; in general entropy differences do not, except if we are subtracting the entropy for a Markov state. Therefore, the entropy difference in de Sitter defined in (13) also satisfies the SSA,

$$\Delta S(A) + \Delta S(B) \geq \Delta S(A \cap B) + \Delta S(A \cup B). \tag{58}$$

Furthermore, since on a null Cauchy surface $S_{\text{rel}} = -\Delta S$, the relative entropy satisfies the strong supperadditivity inequality

$$S_{\text{rel}}(A) + S_{\text{rel}}(B) \geq S_{\text{rel}}(A \cap B) + S_{\text{rel}}(A \cup B). \tag{59}$$

Following the steps of Sec. 4, both $\Delta S$ and $S_{\text{rel}}$ also satisfy the differential versions of these inequalities,

$$\frac{1}{2} \sin(2\theta_0) \partial_{\theta_0}^2 \Delta S(\theta_0) + \partial_{\theta_0} \Delta S(\theta_0) \leq 0,$$
$$\frac{1}{2} \sin(2\theta_0) \partial_{\theta_0}^2 S_{\text{rel}}(\theta_0) + \partial_{\theta_0} S_{\text{rel}}(\theta_0) \geq 0. \tag{60}$$

In order to derive the C-theorem, we note that the running C-function in terms of $\Delta S$ becomes

$$\Delta C(\theta_0) = 3 \tan \theta_0 \, \partial_{\theta_0} \Delta S(\theta_0), \tag{61}$$

and (60) is equivalent to

$$\partial_{\theta_0} \Delta C(\theta_0) \leq 0. \tag{62}$$

This shows that the entropic C-function is monotonically decreasing along the RG flow, providing a proof of the strong version of the C-theorem in de Sitter. It implies the weak version found in Sec. 3.

Let us compare this with the C-theorem in Minkowski space-time. One can recover the flat-space limit by taking the de Sitter radius to be much larger than the size of the entangling region and the RG scale, $\ell \gg L$, $\ell \gg m^{-1}$. Recalling that $L = 2\ell \theta_0$, $L \ll \ell$ implies that $\sin \theta_0 \approx L/2\ell$, and (60) reduces to

$$L \Delta S''(L) + \Delta S'(L) \leq 0, \tag{63}$$

the infinitesimal boosted SSA inequality in flat-space [1]. This inequality implies the monotonicity of the flat space C-function $\Delta C_{\text{flat}}(L) = 3L \Delta S'(L)$, which coincides with the limit $\theta_0 \to 0$ of (61).

As the length $L$ increases, the geometry of de Sitter becomes important for the RG flow, and renders the C-function very different from the flat-space one. As we have emphasized earlier, an important difference is that in de Sitter with fixed $\ell$ we cannot access the flat-space IR limit since we are limited to $L \leq \pi \ell$ (or $\theta_0 \leq \pi/2$). In contrast to what happens in flat space, $\Delta C(\theta_0 = \pi/2)$ is not a difference of UV and IR central charges, but rather depends on the RG flow. We will now turn to a more detailed analysis of $\Delta C(\pi/2)$.

## 5.2 C-function and stress-tensor correlators

In Sec. 3.3 we learned that the quantity $\Delta S(\theta_0 = \pi/2)$ has an information-theoretic interpretation in terms of the distinguishability between the state of the QFT and that of the CFT, both reduced to the static patch. Furthermore, it also has a thermal interpretation since at $\theta_0 = \pi/2$ the EE becomes the thermal entropy. In this section we will show that when the entangling region is the full static patch, $\Delta C(\pi/2)$ can be computed as an integral involving the two-point function of the trace of the stress-tensor.

We work with Euclidean coordinates

$$ds^2 = \frac{\ell^2}{\cosh^2 \tau} \left( d\tau^2 + d\theta^2 \right), \tag{64}$$

and evaluate the derivative of $\Delta S$ from its first variation under metric changes:

$$\delta(\Delta S) = \frac{1}{2} \int d^2x \ \sqrt{g} \, \delta g_{\mu\nu}(x) \langle T^{\mu\nu}(x) \Delta H \rangle. \tag{65}$$

Here $\Delta H = H_\rho - H_\sigma$ is the difference in modular Hamiltonians of the entanglement region for the states $\sigma$ and $\rho$, and the integral runs through the whole space except for small disks around its boundaries, which come from substracting the UV contribution.[9] The CFT modular Hamiltonian $H_\sigma$ was discussed in Sec. 3.1; in contrast, $H_\rho$ is in general nonlocal and not known explicitly.

We want to evaluate the first derivative $\partial_{\theta_0} \Delta S(\theta_0)$ under a change in the size of the causal diamond. We can instead keep $\theta_0$ fixed, but perform an infinitesimal dilation of the euclidean global coordinates,

$$\begin{aligned} \tau &\to (1 + \delta\lambda)\, \tau, \\ \theta &\to (1 + \delta\lambda)\, \theta. \end{aligned} \tag{66}$$

Denoting the corresponding diffemorphism transformation by $\xi_\mu$, we have to evaluate (65) with $\delta g_{\mu\nu} = \nabla_\mu \xi_\nu + \nabla_\nu \xi_\mu$. Note that this transformation is simply a reparametrization of the hyperboloid coordinates, and it does not change its radius $\ell$. A short calculation gives

$$\delta g_{\mu\nu} = \Omega(\tau) g_{\mu\nu}, \quad \Omega(\tau) = 2(1 - \tau \th \tau) \delta\lambda, \tag{67}$$

and thus replacing in (65) we arrive at

$$\partial_\lambda(\Delta S) = \int d^2x \ \sqrt{g} \, \Omega(\tau) \langle \Theta(x) \Delta H \rangle, \tag{68}$$

where $\Theta = g_{\mu\nu} T^{\mu\nu}$. Writing $\lambda = \log(\sin\theta_0)$, the last equation yields an expression for $\Delta C(\theta_0)$.

We are interested in evaluating (68) at the $dS$ horizon $\theta_0 = \pi/2$. This is an important simplification, because the modular Hamiltonian for a QFT is known: it generates time-translations $\partial_t$ preserving the static patch. In terms of global coordinates, and choosing a Cauchy slice at $\tau = 0$, this is

$$H = -2\pi\ell^2 \int_{-\pi/2}^{\pi/2} d\theta \ \cos\theta \, T_{\tau\tau}(0, \theta). \tag{69}$$

This can be obtained by replacing $\theta_0 = \pi/2$ in (29) and (28), followed by a Wick rotation. Then (68) becomes

$$\Delta C(\pi/2) = -6\pi\ell^2 \int_{-\infty}^{\infty} d\tau \int_{-\pi}^{\pi} d\theta \ \sqrt{g} \, \Omega(\tau) \int_{-\pi/2}^{\pi/2} d\theta' \ \cos\theta' \langle \Theta(\tau, \theta) \Delta T_{\tau\tau}(0, \theta') \rangle. \tag{70}$$

---

[9]One can arrive at expression (65) by writing the reduced density matrix as a path integral and using the first law of entanglement $\delta S = \mathrm{tr}(\delta\rho H)$, similarly as in [39].

In order to evaluate this quantity, we will use the spectral representation for the stress tensor 2-point function in de Sitter. The necessary tools have been recently developed, see e.g. [15, 16]. These methods were applied to prove the C-theorem using correlators in [17], and we discuss the connection with these works below. The correlator of two stress-tensor operators in $dS$ can be written as a certain integral running trough the unitary irreducible representations of $SO(1,2)$,

$$\langle T_{\mu\nu}(x)T_{\alpha\beta}(x')\rangle = \int_\Delta \frac{\varrho(\Delta)}{[2+\Delta(1-\Delta)]^2} \Pi_{\mu\nu}(x)\Pi_{\alpha\beta}(x') G_\Delta(x,x'). \tag{71}$$

Here $\Pi_{\mu\nu}(x)$ is the differential operator given by

$$\Pi_{\mu\nu}(x) = g_{\mu\nu}(x)(\nabla^2+1)-\nabla_\mu\partial_\mu, \tag{72}$$

which ensures the Ward and trace identities for the stress tensor [40]. $G_\Delta(x,x')$ is the Green's function for a free scalar field, which satisfies

$$\nabla^2 G_\Delta(x,x') = \Delta(1-\Delta)G_\Delta(x,x'), \tag{73}$$

away from coincident points. And $\varrho$ is the spectral density for the stress-tensor, which receives contributions from the principal and complementary series (parametrized by some $\Delta \in 1/2+i\mathbb{R}$ and $\Delta \in (0,1)$, respectively), and possibly also a discrete series contribution. These details will not be needed for our discussion, but further information may be found in [17].[10]

Replacing (71) into (70) and integrating by parts twice with respect to $\theta'$, we have

$$\Delta C(\pi/2) = -6\pi\ell^2 \int_\Delta \frac{\varrho(\Delta)}{2+\Delta(1-\Delta)}$$
$$\times \int_{-\infty}^\infty d\tau \int_{-\pi}^\pi d\theta \ \sqrt{g}\,\Omega(\tau)\left[G_\Delta(\tau,\theta;0,\pi/2)+G_\Delta(\tau,\theta;0,-\pi/2)\right]. \tag{74}$$

We discard coincidence-point contributions because they cancel out in the subtraction $\Delta C$ and $\Delta T_{\tau\tau}$. In order to further simplify this expression, we use that $G_\Delta$ is actually a function of the geodesic distance between $x$ and $x'$, which can be expressed as

$$L = \ell\arccos\sigma, \quad \sigma = \frac{\cos(\theta-\theta')}{\cosh\tau\cosh\tau'} - \text{th}\,\tau\,\text{th}\,\tau'. \tag{75}$$

When one of the points is evaluated at the horizon, i.e. $\tau' = 0$ and $\theta' = \pm\pi/2$, we have $\sigma = \pm\sin\theta/\cosh\tau$, ranging from 1 to $-1$ for coincident and antipodal points, respectively. We can use this variable in (74) instead of $\theta$, so one arrives at

$$\Delta C(\pi/2) = -12\pi\ell^4 \int_\Delta \frac{\varrho(\Delta)}{2+\Delta(1-\Delta)}$$
$$\times \int_{-1}^1 d\sigma \int_{-\cosh^{-1}\frac{1}{|\sigma|}}^{\cosh^{-1}\frac{1}{|\sigma|}} d\tau \ \frac{1-\tau\,\text{th}\,\tau}{\cosh\tau\sqrt{1-\sigma^2\cosh\tau}} \left[G_\Delta(\sigma)+G_\Delta(-\sigma)\right]. \tag{76}$$

Combining both Green's functions $G_\Delta(\sigma)+G_\Delta(-\sigma) \to 2G_\Delta(\sigma)$, and performing the $\tau$ integral, we arrive at

$$\Delta C(\pi/2) = -24\pi^2\ell^4 \int_\Delta \frac{\varrho(\Delta)}{2+\Delta(1-\Delta)} \int_{-1}^1 d\sigma \ |\sigma|G_\Delta(\sigma). \tag{77}$$

---

[10]Note that the parameter $\Delta$ introduced here is not related to the dimension of the primary operators that appear at (10) which we had called $\Delta_I$.

One can check that $|\sigma| = (\nabla^2 + 2)f(\sigma)$ for the positive function

$$f(\sigma) = \frac{1}{3}\left[1 - \sigma - \sigma \log\left(\frac{1+\sigma}{2}\right)\right] + \frac{1}{3}\theta(-\sigma)\sigma \log(1-\sigma^2) \tag{78}$$

(with $\theta(x)$ the Heaviside step-function), which is continuous and differentiable at $\sigma = 0$.[11] Replacing this, integrating by parts, and using the equation of motion for $G_\Delta$ once again, we arrive at

$$\Delta C(\pi/2) = -24\pi^2 \ell^4 \int_{-1}^{1} d\sigma\, f(\sigma) \int_\Delta \varrho(\Delta) G_\Delta(\sigma). \tag{79}$$

Finally, using again (71) to identify $\int_\Delta \varrho(\Delta) G_\Delta(\sigma) = \langle\Theta\Theta\rangle(\sigma)$ we obtain

$$\Delta C(\pi/2) = -24\pi^2 \ell^4 \int_{-1}^{1} d\sigma\, f(\sigma) \langle\Theta\Theta\rangle(\sigma). \tag{80}$$

This is our final result for $\Delta C(\pi/2)$ in terms of stress-tensor correlators. We see that $\Delta C(\pi/2) < 0$, which we derived before using monotonicity of the relative entropy, or the SSA, is equivalent to the positivity of the stress-tensor trace 2-point function on the sphere.

For a QFT in flat space, a conceptually similar (but technically simpler) analysis leads to the relation [8, 27, 41]

$$\Delta C_{\text{flat}} = -3\pi \int d^2x\, x^2 \langle\Theta(x)\Theta(0)\rangle. \tag{81}$$

This is a sum rule, in the sense that the integral over all the Euclidean plane in the right hand side should give a quantity $C_{\text{IR}} - C_{\text{UV}}$ that depends only on the fixed points. We have arrived at a similarly-looking formula (80) for $dS$, with some key differences. First, this is no longer a sum rule, but in general it depends on the details of the RG. Secondly, the kernel $f(\sigma)$ takes into account the effects of the nontrivial $dS$ geometry. We can relate the two formulas by taking a limit of small separations in $dS$. Indeed, for a geodesic separation $L \ll \ell$, (75) gives

$$\sigma \approx 1 - \frac{L^2}{2\ell^2}, \tag{82}$$

and

$$f(\sigma) \approx \frac{L^2}{4\ell^2}. \tag{83}$$

Then the regime of the integral (80) near $\sigma = 1$ reduces to the flat space result (81), after performing the angular integration.

## 5.3 Comparison with unitarity methods

Recently, [17] proposed a C-theorem for de Sitter using unitarity methods. Let us briefly compare the two approaches, with a view towards establishing a more direct connection between information theory and unitarity results.

The basic idea is to extend to de Sitter the flat-space sum rule for the C-theorem. In this approach, there is no entangling region whose size can be used to probe the RG, but instead the $dS$ radius $\ell$ is varied: $m\ell \ll 1$ accesses the UV, and $m\ell \gg 1$ probes the IR. The proposed C-function is of the form [17]

$$\Delta \tilde{C}(\ell) = -\int_{-1}^{1} d\sigma\, r(\sigma) \langle\Theta\Theta\rangle(\sigma), \tag{84}$$

---

[11]There are several functions $f(\sigma)$ satisfying $(\nabla^2+2)f(\sigma) = |\sigma|$, but the one given in (78) is fixed by demanding $f(\sigma = 1) = 0$, in order to eliminate the divergence of $G_\Delta$ in the coincidence limit; and regularity at $\sigma = 0$, in order to cancel the boundary terms appearing while integrating by parts.

and $r(\sigma)$ is chosen such that

$$r(\sigma)\langle\Theta\Theta\rangle(\sigma) = \frac{\mathrm{d}}{\mathrm{d}\sigma}\tilde{C}(\sigma).\tag{85}$$

Then the integral localizes on the endpoints, and one has to require that at coincidence points, $\tilde{C}(\sigma = 1) = C_{\mathrm{UV}}$. The C-function corresponds then to the value of $\tilde{C}$ at antipodal points, $c_1(\ell) = \tilde{C}(\sigma = -1)$. Ref. [17] found

$$r(\sigma) = 8\pi^2\ell^4\left[1 - \sigma - \sigma\log\left(\frac{1+\sigma}{2}\right)\right].\tag{86}$$

Interestingly, $r(\sigma)$ coincides with the entropic kernel $f(\sigma)$ that we found in (78) for $\sigma > 0$. However, both are different for $\sigma < 0$. In our case, $f(\sigma)$ is determined by the modular Hamiltonian for the static patch, while the proposal by [17] follows from requiring (85), which our function does not satisfy. Another important difference is that $f(\sigma) > 0$, while $r(\sigma)$ becomes negative and divergent near $\sigma = -1$. Despite this lack of positivity, and given a plausible assumption on the absence of discrete series contributions, unitarity constraints give $\Delta\tilde{C} \leq 0$ [17].

Of course, there is no contradiction in having different running C-functions, something that also happens in the context of Zamolodchikov's C-theorem. But the equality between $f(\sigma)$ and $r(\sigma)$ suggest a deeper connection between the two approaches. Both give the same IR or flat space limit,

$$\lim_{\ell\to\infty}\Delta C(\pi/2) = \lim_{\ell\to\infty}\Delta\tilde{C} = C_{\mathrm{IR}} - C_{\mathrm{UV}}.\tag{87}$$

In particular, it is well-motivated to conjecture that

$$\frac{\mathrm{d}}{\mathrm{d}\ell}\Delta C(\pi/2) \leq 0,\tag{88}$$

so that the value of the entropic C-function for the static patch, $C(\pi/2)$, monotonically interpolates between $C_{\mathrm{UV}}$ and $C_{\mathrm{IR}}$. As we discussed before, changing the $dS$ radius requires additional work in the information-theoretic framework, and this is a direction where the different methods can fruitfully intersect.

# 6 Conclusions and future directions

In this work we proved the irreversibility of the renormalization group in two-dimensional de Sitter spacetime using quantum-information methods. We constructed a monotonic C-function given in terms of the derivative of the entanglement entropy difference between the Euclidean vacuum state of the UV CFT (reduced to an interval), and that of the QFT that undergoes the RG flow. For large de Sitter radius our entropic C-function reduces to the Minkowski one [1], but in general it is quite different. Keeping the $dS$ radius fixed while varying the length of the interval allows us to probe the same QFT at various scales. For short scales the C-function gives the central charge of the UV theory. The minimum value of our C-function is attained when the entangling region covers the whole static patch; this value depends on the RG flow.

We demonstrated the irreversibility using two independent, fully Lorentzian approaches. The first one is based on the monotonicity of the relative entropy, and yields a weak version of the C-theorem (in the sense that it gives a bound for our C-function in terms of the central charge of the UV fixed point). We showed that the entropy difference between the state of the CFT and that of the QFT reduced to the causal diamond coincides with (minus) the corresponding relative entropy when evaluated at the null surface. From the positivity of the relative entropy, we also obtained a bound on the entanglement and thermal entropies for a QFT on de Sitter.

The second approach uses the strong subadditivity property of the entanglement entropy for boosted regions and the Markov property of the CFT vacuum. Using a geometrical setup analogous to the one involved in the proof of the entropic C-theorem for Minkoski spacetime, we showed that the C-function for de Sitter has non-positive derivative and thus decreases monotonically with the length of the interval until it reaches the horizon of the static patch. The isometries of de Sitter play an important role in this derivation.

Finally, we also computed an expression for the C-function evaluated at the static patch horizon in terms of correlators of the stress-tensor. In the flat-space limit, our result reduces to the sum rule for the difference between the UV and IR central charges. Our formula appears to be closely related to the one presented in [17], using unitarity methods.

## 6.1 Future directions

There are several directions in which this work could be followed, as well as some open questions that we wish to answer in the future. Let us name a few of them.

- It would be interesting to calculate our running C-function for simple models on the lattice in order to gain more insight on its behaviour. We anticipate that this will require to develop some suitable discretization algorithm for an expanding universe such as de Sitter.

- We restricted to regions inside the static patch of de Sitter in this manuscript, and this renders the entanglement entropy of such regions time-independent. Instead, some works (see for instance [26]) focus on computing super-horizon entropies, which are time-dependent. We would like to make contact with this approach in the future.

- Since we worked with a fixed radius $\ell$, our C-function interpolates between the UV central charge $C_{\text{UV}}$ and $C(\pi/2)$, a quantity that depends on the RG flow. In this work we have proven that $C_{\text{UV}} \geq C(\pi/2)$. Also, it is broadly known that $C_{\text{UV}} \geq C_{\text{IR}}$, but it is still not clear if the relation $C_{\text{UV}} \geq C(\pi/2) \geq C_{\text{IR}}$ holds, although we expect it does. In order to show its validity we would need to study the dependence of $C(\pi/2)$ with $\ell$, particularly the limit $\ell \to \infty$, in order to probe the IR.

- The quantum information methods we employed in this paper can also be applied to higher-dimensional space-times [19]. In particular, this would give the F and A theorems in de Sitter. More generally, we think that the entropic methods could also apply to other curved space-times with less symmetries (e.g. by incorporating defects) or to QFTs in anti-de Sitter (AdS). This may have interesting connections with quantum gravity, such as gaining insight on the logarithmic corrections to the Bekenstein-Hawking formula for the entropy of black holes in AdS.

## Acknowledgments

We thank H. Casini and M. Huerta for multiple discussions on the subject.

**Funding information**  NA is supported by a CONICET PhD fellowhip. GT is supported by CONICET (PIP grant 11220200101008CO), CNEA, and Instituto Balseiro, Universidad Nacional de Cuyo.

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
