# Peer review of "Quantum information and the C-theorem in de Sitter"

_SciPost Physics, doi:SciPost Phys. 18, 029 (2025)_

## Round 1 · Referee Report · Anonymous (Referee 1) · 2024-11-19

Strengths

1 - Interesting application of quantum information methods to QFT in de Sitter spacetime

Weaknesses

1- Lack of examples.

Report

This paper studies entanglement and relative entropies in QFT in de Sitter spacetime. In particular, it proves a monotonicity theorem for a C-function - equation (5.6). In addition, equation (5.24) relates this C-function to the two-point function of the trace of the stress tensor. These are interesting results that justify publication.

Requested changes

1- after equation (2.13) it is written: "In general dimensions, it is possible to supplement the UV fixed point with appropriate counterterms in order to render ∆S(V ) finite." Why? Is there a simple argument? Can you provide references?

2- Equation (3.1) assumes that the density matrices $\sigma$ and $\rho$ act on the same Hilbert space. This is not obvious because they correspond to different QFTs.

3- In equation (3.9), why is $k$ quadratic in the couplings $\lambda_I$?

4- The log divergence in (3.11) should be treated more carefully. I suppose one should study the region of small $\theta$ before expanding at small $a$.

5- Is it possible to express $\Delta C(\theta_0)$ in terms of the stress tensor two-point function? Or this is only possible for $\theta_0=\pi/2$ like in equation (5.24)?

6- It would be instructive to compute the proposed C-function in some examples, like the theory of a massive free boson or a massive free fermion.

Recommendation

Ask for minor revision

---

## Round 1 · Referee Report · Anonymous (Referee 2) · 2024-11-22

Report

Finding quantities that decrease monotonically along the renormalization group (RG) flow is an important problem in quantum field theory (QFT). For QFT in flat spacetime, such quantities have been identified in various dimensions, such as the central charge in 2d, the F-function in 3d, and the a-anomaly in 3d. In particular, the monotonicity of the central charge in 2d is known as the C theorem. This paper takes the important step of extending the C theorem to 1+1 dimensional de Sitter spacetime using an information-theoretic method. This is an intriguing result with potential applications in cosmology. However, I have some comments below that I believe need to be addressed before I can support the paper for publication.

Introduction:

The introduction is relatively short. It would be beneficial if the authors could briefly review and discuss the following points: 1. What tools from information theory are used, and how do they apply in flat spacetime? 2. What are the main difficulties when working with dS compared to flat spacetime? 3. What is the advantage of using the information-theoretic method in dS compared to the unitarity method?

Apart from these general remarks, there are two more specific comments:

  1. It's mentioned " much less is understood about theories without Lorentz invariance". But dS has Lorentz symmetry. Do the authors mean Poincare symmetry ?

  2. The authors cited papers on c-theorem [7] and a-theorem [9]. It makes more sense to also cite works on F-theorem, e.g.
    "Towards the F-Theorem: N=2 Field Theories on the Three-Sphere" and "F-Theorem without Supersymmetry".

Section 2:

  1. Equation (2.23) does not necessarily imply that "it attains its minimum negative value for "\theta_0=π/2." It only indicates that the first derivative of ΔS vanishes at this point. How do the authors rule out the possibility of a local maximum instead?

Section 3:

1.I believe equation (3.1) requires more explanation, particularly regarding the meaning of the trace. Since \rho and \sigma are density matrices in different theories, they act on different Hilbert spaces. Over which Hilbert space is the trace "\tr" taken in this case?

  1. Below equation (3.11), it is mentioned that the logarithmic divergence is suppressed by a^2. It seems that the authors have exchanged the order of integration and the limit a->0. Could the authors please comment on this operation?

  2. S_{\thermal} in equation (3.15) is supposed to be regularized. Could the authors specify the regularization prescription used? Is this result independent of the regularization scheme?

Recommendation

Ask for minor revision

---

## Round 2 · Referee Report · Anonymous (Referee 2) · 2024-12-9

Report

The Authors replied positively to the comments of the previous report. It can be published in the present form.

Recommendation

Publish (meets expectations and criteria for this Journal)

---

## Round 2 · Referee Report · Anonymous (Referee 1) · 2024-12-11

Report

I thank the authors for addressing my questions. I recommend the paper for publication.

Recommendation

Publish (easily meets expectations and criteria for this Journal; among top 50%)

---

## Round 2 · Author Response

We thank the referees for their detailed and helpful reviews. In the new version of the manuscript, we have addressed their points. We next detail a list of changes.

---

## Round 2 · List of Changes

REPORT 1

  1. We have added Footnote 4 to explain this point.
  2. This is an important point, also raised by the other referee. We have added a new paragraph on this at the beginning of Sec. 3. We have explained this point below Eq. (3.8).
  3. This point was also raised by the other referee. We have added a short distance cutoff to regularize the integral of eq. (3.11) and expanded on this below such equation.
  4. In the evaluation of (5.24) we used the fact that the QFT modular hamiltonian is known to be the generator of static time evolution. This is a key simplification since for arbitrary regions the modular hamiltonian for the QFT is not local, and there does not exist a simple expression in terms of correlators for $\Delta C(\theta_0)$.
  5. We agree that providing examples in free massive theories would be very useful. We are currently working on this project, which turns out to be nontrivial due to the de Sitter scale factor.

REPORT 2

Introduction 1.2.3. Added 3 new paragraphs to the introduction in order to address the questions in the referee report. 4. Replaced Lorentz invariance by Poincare invariance in the Introduction 5. Added refs. on the F-theorem in the Introduction.

Section 2 1. We explained below (2.23) why the extremum is a minimum.

Section 3 1. This is an important point and we agree that it requires more explanations. We have added a new paragraph on this at the beginning of Sec. 3.

  1. We have expanded on this point below eq. (3.11).

  2. We have clarified this point below. eq. (3.15).

We also note that we have fixed a typo in a factor of 2 in Eq. (4.10).

---

## Editorial Decision

published